# Revealing the hidden third dimension of point defects in two-dimensional MXenes

Grace Guinan [1], Michelle A. Smeaton [1], Brian C. Wyatt [2,3], Steven Goldy [4], Hilary Egan [1], Andrew Glaws [1], Garritt J. Tucker [5], Babak Anasori [2,6] & Steven R. Spurgeon [1,4,7] ✉

Point defects govern many important functional properties of two-dimensional (2D) materials. However, resolving the three-dimensional (3D) arrangement of these defects in multi-layer 2D materials remains a fundamental challenge, hindering rational defect engineering. Here, we overcome this limitation using an artificial intelligence-guided electron microscopy workflow to map the 3D topology and clustering of atomic vacancies in $Ti_3C_2T_X$ MXene. Our approach reconstructs the 3D coordinates of vacancies across hundreds of thousands of lattice sites, generating robust statistical insight into their distribution that can be correlated with specific synthesis pathways. This large-scale data enables us to classify a hierarchy of defect structures—from isolated vacancies to nanopores—revealing their preferred formation and interaction mechanisms, as corroborated by molecular dynamics simulations. This work provides a generalizable framework for understanding and ultimately controlling point defects across large volumes, paving the way for the rational design of defect-engineered functional 2D materials.

Two-dimensional (2D) materials have become a major field of modern research in materials science since the discovery of graphene in 2004. Since then, applications of 2D materials have spanned from energy storage and conversion, biomedical, sensors, microelectronics, or mechanical reinforcements[1–5]. Typically, the design of 2D materials for different functionalities requires scientists to precisely control composition, structure, and surface chemistry at the atomic level[6–8]. Atomic-level point defects, especially vacancies, can significantly alter the electronic, electrochemical, catalytic, mechanical properties, and performance of 2D materials[9–14]. Despite knowledge of the detrimental effects of vacancies on the properties of 2D materials, characterizing their presence and distribution remains a challenge using conventional techniques. The challenge of characterizing point defects is significantly compounded in few-layered 2D materials. For instance, MXenes—a class of 2D transition metal carbides, carbonitrides, and nitrides—consist of nanosheets containing two to five layers of metal atoms, which complicates defect analysis compared to single-layer materials.

To characterize atomistic defects, researchers traditionally use atomically-resolved electron microscopy, such as aberration-corrected scanning transmission electron microscopy (STEM). This technique enables highly detailed visualization and analysis of atomic-level defects and resulting structural changes[15–18]. For most researchers working on quantifying defects in 2D materials, this process involves manual identification in low-dose conditions, which greatly limits both the volume of material that can be studied and the accuracy of any resulting measurements, which could lead to statistically unsound conclusions[18]. This gap in our knowledge of 3D defect populations is therefore a fundamental barrier to understanding and control of defect-property relationships in multi-layered 2D materials.

[1]National Laboratory of the Rockies, Golden, CO, USA. [2]School of Materials Engineering, Purdue University, West Lafayette, IN, USA. [3]Applied Materials Division, Argonne National Laboratory, Lemont, IN, USA. [4]Metallurgical and Materials Engineering Department, Colorado School of Mines, Golden, CO, USA. [5]Department of Physics and Astronomy, Materials Science Program, Baylor University, Waco, TX, USA. [6]School of Mechanical Engineering, Purdue University, West Lafayette, IN, USA. [7]Renewable and Sustainable Energy Institute, University of Colorado Boulder, Boulder, CO, USA. ✉e-mail: steven.spurgeon@nlr.gov

Over the past decade, artificial intelligence (AI) and machine learning (ML) methods have emerged as a path to robust statistical analysis across many scientific disciplines, ranging from medicine to chemical synthesis. These methods have begun to transform the analysis of atomic-resolution microscopy data[19,20], enabling high-throughput identification and classification of point defects in monolayer 2D materials such as graphene[20,21] and transition metal dichalcogenides (TMDs)[22]. Nonetheless, it remains a significant challenge to extend these powerful techniques to multi-layer systems, since we must deconvolve 3D structural information from 2D projection images at scale. A recent pioneering study[23] demonstrated an ML workflow to identify and quantify vertically stacked bivacancies in bilayer CrSBr, representing a critical first step towards multi-layer defect analysis. However, a full 3D topological analysis of defect clustering in systems with three or more atomic layers, which allows for a richer hierarchy of complex, multi-layer defect motifs, has remained elusive. Here, we overcome this limitation by developing an AI-guided electron microscopy framework to map the 3D topology of atomic vacancies across all three metal layers of $Ti_3C_2T_x$ MXene.

MXenes have attracted immense research interest, comprising a substantial fraction of all publications on 2D materials in the first half of 2025, according to the Web of Science. This attention is in part due to their usefulness across a variety of fields, including advanced electronics, water purification, energy storage, and biomedicine and their compositional tunability[24–26]. MXenes are denoted by their chemical formula $M_{n+1}X_nT_x$, where M is comprised of $n+1$ layers of one or more early transition metals, X is C and/or N, and $T_x$ represents surface terminations bonded to the exterior transition metal planes[27–29]. MXenes represent an ideal material class to demonstrate advancements in defect analysis for several key reasons: their diverse chemistries and structures, controllable defect generation, prominence in 2D materials research, and broad applicability across science and engineering.

Traditionally, MXenes are synthesized using aqueous-based acids, commonly hydrofluoric acid, to selectively etch them from their precursor MAX phases[27]. As a result of this harsh etching process, transition metal vacancies can be induced on the freshly exposed surface transition metal planes. It is known that the concentration of these vacancies can be controlled by tuning the acid concentration during synthesis[26,30]. However, the role of acid content on the distribution and

clustering behavior of these vacancies remains a major gap in MXene research despite its importance in MXenes' material properties and stability. A major reason for this current gap is that, so far, imaging techniques have been limited in their ability to explicitly quantify the link between processing and specific 3D defect distributions within the MXenes, owing to the challenges of manual analysis. While ML has been applied to MXenes for high-throughput computational screening and property prediction[31], the large-scale statistical analysis of their 3D defect structures from experimental data and the direct correlation with synthesis pathways is a major gap in the field.

In this work, we address this critical need by introducing an AI-guided STEM framework to perform a large-scale statistical analysis of 3D point defect complexes in multi-layered 2D materials, using $Ti_3C_2T_x$ MXene as a model system. We introduce an AI-guided STEM approach to classify and model point defect clustering in each individual MXene metal plane, observing over 3000 titanium vacancies across 150,000 atomic lattice sites. We find that using harsher selective etching conditions (i.e., increasing the concentration of hydrofluoric acid in MXenes' synthesis), results in increased vacancy formation in both the surface and subsurface transition metal planes in MXenes. Further, we observe that increasing the etching-induced vacancy formation typically yields an increased concentration of both vacancy clusters and nanopore formation in individual 2D sheets of MXenes, as compared to individual point vacancies. This integration of theory and experiment, facilitated by statistics and machine learning, gives a more complete understanding of defective MXenes' structure, which can lead to higher fidelity modeling and prediction of material performance for future critical applications like mechanical behavior or energy conversion. Overall, this work demonstrates how AI-guided characterization can transform the identification and statistical analysis of 3D point defect distributions in multi-layer 2D materials.

## Results

### An AI-guided workflow for 3D vacancy mapping

We synthesized variable concentrations of defects on $Ti_3C_2T_x$ MXenes by controlling the concentration of hydrofluoric acid (HF) solution in a mixed HF and hydrochloric acid (HCl) etchant solution (5, 9.1, or 12.5% HF) (Fig. 1a), similar to previous work[30]. We chose this approach as we hypothesized that controlling the HF concentration would induce a higher concentration and clustering of defects, as well as potential

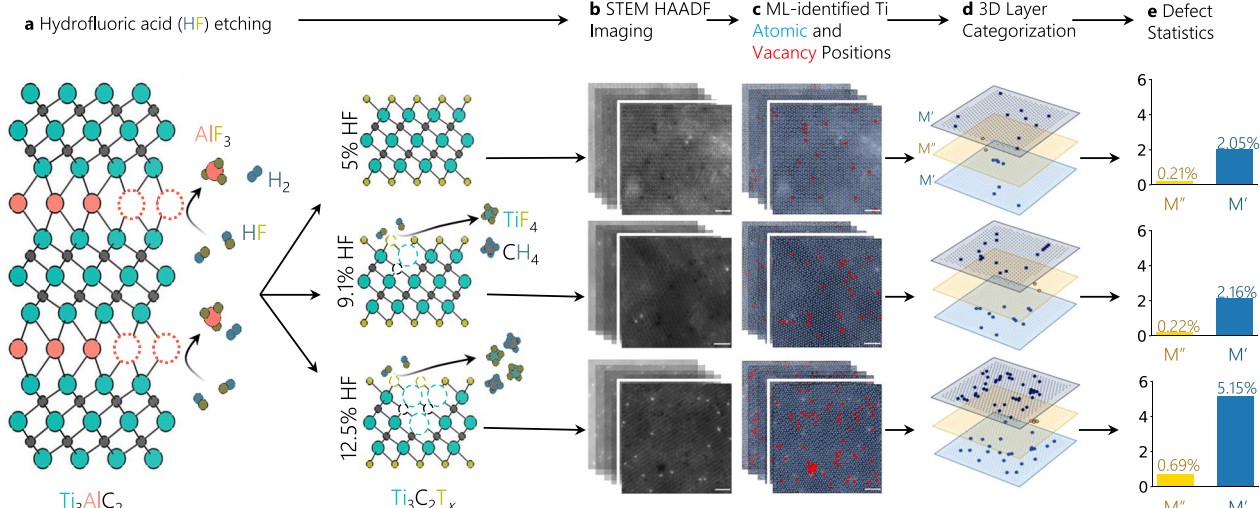

**Fig. 1 | AI-guided microscopy analysis of 3D point defect distributions in MXenes. a** Approach to synthesize variably defective $Ti_3C_2T_x$ MXenes using aqueous HF etching approaches from the $Ti_3AlC_2$ MAX phase; model labels are color coded to the atomic sphere colors. **b** STEM images (scale bar 1 nm) of variably defective $Ti_3C_2T_x$ MXenes, with (**c**) overlaid AI-guided identification of atoms (blue)

and defects (red) on the STEM images. **d** The AI-guided identification of defects is further broken down by their occupancy M' (blue) and M" (orange) layers with bold vacancies (**e**) Averaged vacancy percentage in the M' and M" layers depending on HF content used in synthesis. Y axis is percent vacancies.

damage to the interior structure of the flakes of 2D MXenes, as discussed previously[26,30]. Further details on the etching conditions and preparation can be found in the Synthesis section of the Methods. We imaged these samples using STEM to obtain high-angle angular dark-field (HAADF) images of monolayer flakes in each MXene sample, as described in the Methods and shown in Fig. 1b. STEM imaging allows us to visualize, and later classify, individual atoms and defects; however, traditional imaging is challenging due to the extreme beam sensitivity of MXenes to knock-on damage[32,33]. To mitigate this, both a 60 kV accelerating voltage and low (10–25 pA) beam current were used. We designed an ML-based classification approach that demonstrates superior performance in atom and defect identification from low signal-to-noise ratio images compared to conventional Gaussian fitting methods.

To build our statistical model of the generated defects, we conducted a bootstrapping analysis to understand the statistical spread of our data and to compare the vacancy percentage of our different HF samples. We took 28 images of the 5% sample, 23 of the 9.1%, and 26 of the 12.5% HF $Ti_3C_2T_x$ MXenes. This data was sufficient to statistically differentiate the vacancy percent of the 12.5% sample from the 5% and 9.1%; however, the 5% and 9.1% samples were not found to be statistically different (further details in Supplementary Fig. S5).

After determining the number of images required to gain statistical confidence in our defect averages, we employed ML to identify atom and defect positions in our images using a neural network for semantic segmentation of our images followed by clustering for atom and vacancy instance segmentation (details in Methods and Supplementary Note 1). Our neural network framework identified over 150,000 atoms and 3000 vacancies across the three Ti atomic planes, demonstrating the high-throughput capability essential for robust statistical analysis (Fig. 1c). When comparing vacancy concentrations across samples, we found 3.49% defects in the 12.5% sample compared to 1.48% and 1.41% defects in the 9.1% and 5% samples, respectively. Since defects make up only a small percentage of the material (1–4 % of atomic positions), the accuracy of our model was important. However, the strong effect of imaging conditions on the representation of objects in electron microscopy leads to a lack of labeled, experimental ground truth data; this makes it difficult to quantitatively evaluate model performance in a traditional sense, as has been described elsewhere[34,35]. Rather, we evaluate our model performance relative to conventional 2D Gaussian fitting on the following criteria: accuracy of determining atomic positions, as evaluated by expert microscopists; the degree of required manual tuning of fitting parameters; and robustness against imaging noise. Using these criteria, our model exhibits far superior performance, as described in Supplementary Note 1.

We then extended this 2D analysis to reconstruct the full 3D defect topology by deconvolving the three constituent Ti layers. ML is particularly powerful here, since it provides all atomic and defect positions. Given all these locations, we took advantage of the positioning of the material with respect to the beam (as seen in Fig. 2b) and the lattice angles associated with the hexagonal structure of the MXenes to determine the relative layer of each atom (this tiling method shown is further detailed in the Machine Learning and Statistical Analyses section of the "Methods" and Fig. S3). Next, we counted the number of vacancies in each relative layer and took the layer with the least number of vacancies to be the middle layer (M") and the other two layers to be outer layers (M'). This assumption is backed up both by previous theoretical studies[36], which found that surface vacancies have a much lower formation energy than inner vacancies, as well as our own findings, which show one layer with far fewer vacancies than the other two layers, across all three HF samples (Fig. 1e).

The differentiation of the middle layer (M") from the outer two layers (M') (Fig. 1d) enabled us to resolve the vacancy concentration in all three transition metal planes, showing an average increase of M' and M" vacancies from 2.05 at% and 0.21 at% in 5 wt% HF, respectively, to 5.15 at% and 0.69 at% in the 12.5 wt% HF samples, respectively (Fig. 1e). These values are comparable to previous studies demonstrating a ~1 at % vacancies in the M' layer for high-quality MXenes[26]. From this foundation, we can quantify the spatial distribution and classify the clustering behavior of vacancies using these newly created 3D models of defects in variably defective MXenes.

## Classification of 3D defect motifs

We used our 3D layer-by-layer modeling of Ti vacancies in $Ti_3C_2T_x$ MXenes, enabled by our ML approach, to define detailed classifications of vacancy arrangements in 2D MXenes (Fig. 2a). This step took us beyond basic statistics of vacancy concentration, allowing us to describe and quantify exquisite detail of vacancy clustering dynamics with processing, such as the clustering commonalities across our whole dataset as the differences between the three different % HF samples. Such information is critical to understand defect clustering and inform theory models for synthesis pathways. Here, ML allows us to analyze a greater quantity of images in more detail than would be feasible by hand, giving us confidence in our higher order statistics and allowing us to define the following defect motifs.

We classified the observed vacancies into four distinct motifs based on their 3D clustering behavior (Fig. 2b): 1) isolated vacancies, which are single point vacancies that are at least one nearest neighbor away from another vacancy; 2) surface clusters, aggregates of vacancies confined to a single metal plane (predominantly observed in the outer M' layers); 3) inter-layer clusters, aggregates of vacancies spanning at least two adjacent metal planes, and 4) nanopores, where the vacancies are present as first nearest neighbors through all three Ti layers, indicating a hole in the flake.

To define these defect types, we used the atomic layer to classify vacancies by their relative positions in the 3D lattice. We used a Delaunay triangulation to define defects that share an edge, meaning they are directly next to one another in the STEM images or directly next to one another in their respective layer (more details on the triangulation are provided in Supplementary Fig. S4). Using these classifications, we further discriminate this data to analyze the frequency of vacancy types present by dividing the number of vacancies in each classification by the total number of vacancies present, as shown in Fig. 2b. We observe that isolated vacancies account for 47.01% of all vacancies, surface clusters for 27.61%, inter-layer defects for 18.90%, and nanopores for 5.85%.

Overall, we designed this study to investigate the effect of increasing the total HF concentration on the total number and clustering behavior of defects in our MXenes, as shown in Fig. 1e. We further hypothesized that if over-etching were to occur due to a higher concentration of HF in the etchant, as suggested in previous works[26,30], it would result in vacancy clustering rather than isolated point vacancies, as atoms around vacancy sites would likely be easier to remove than those with a pristine surface surrounding them. We expand on potential sources of this energetic favorability based on simulations later in Results. However, we next divided the number of cluster vacancies by the total number of vacancies in clusters (types 2–4), as shown in Fig. 2c. In addition, we calculated the size of our vacancy clusters (types 2–4) by counting the total number of vacancies involved in each type by HF content used in its synthesis as shown in Fig. 2d.

We found that vacancy clusters comprised a larger portion of vacancies (Fig. 2b) for the 12.5% HF etched $Ti_3C_2T_x$ MXene, at 59.05%, than 5 or 9.1% HF at 50.33 and 38.29%, respectively. We also found that the proportion of inter-layer and nanopore vacancy clusters (types 3 and 4) generally increased for an increased concentration of HF from 10.09 to 24.72% inter-layer vacancies and 4.12 to 8.29% nanopores for 5% and 12.5%, respectively. The 12.5%-HF etched sample exhibited the

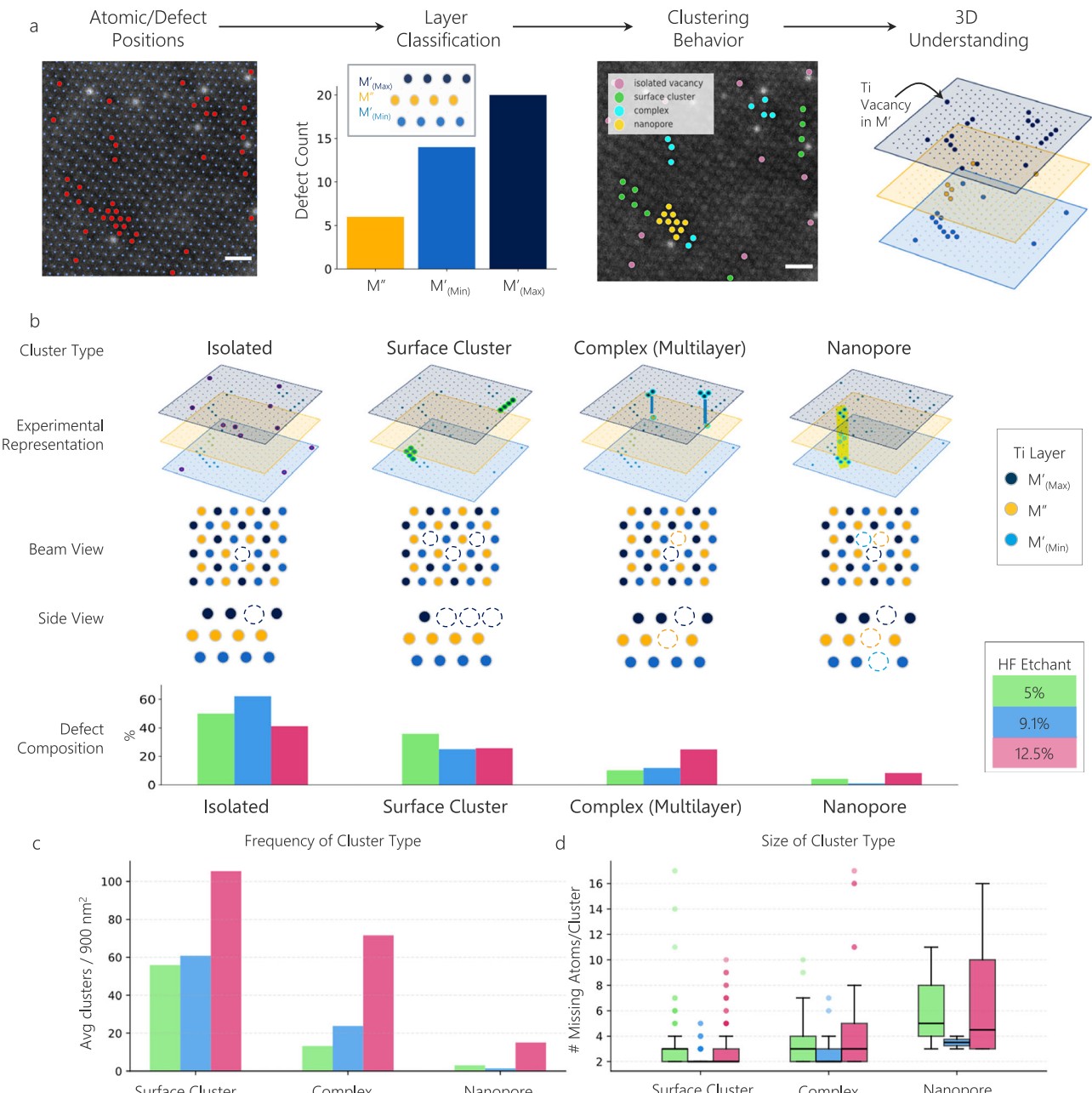

**Fig. 2 | Vacancy types observed in over-etched Ti$_3$C$_2$T$_x$ MXenes. a** Approach used to classify defect types by classifying layers; M" (yellow) vs M' (blue). Scale bars 0.5 nm. **b** Identification of defect types and composition of these types of proportional to all defects found on single flakes of MXenes made with variable concentrations of HF; 5% (green), 9.1% (blue), 12.5% (pink). Model derived from experimental data presented in part (**a**). **c** Frequency of clustering types on MXenes based on HF concentration. **d** Size of the clusters in MXenes based on type and concentration of HF. Boxplots display the median (line), interquartile range (box), 1.5 times the IQR (whiskers), and points (outliers).

highest frequency of surface cluster, inter-layer, and nanopore vacancies compared to the 5%- and 9.1%-HF etched samples (Fig. 2c). However, the average number of atoms per cluster type was relatively similar across the three samples (Fig. 2d). This result indicates that while the 12.5% HF etchant produced more clusters overall, the vacancies did not coalesce into larger defects but remained similar in size to those observed in the 5% and 9.1% cases.

In contrast to our hypothesis, we found that the 9.1% HF etched Ti$_3$C$_2$T$_x$ MXene showed a higher proportion of isolated vacancies, with 61.72%, than the 5% HF Ti$_3$C$_2$T$_x$ MXene, with only 49.67%. Similarly, we observed that the vacancy concentration does not increase (Fig. 1d) as much for 5 to 9.1% HF in M' sites (2.05 to 2.16 at%, respectively) as it did for 9.1% HF to 12.5% HF in M' (2.16% to 5.15%, respectively). When we

analyzed the XRD patterns for the post-etched samples (see Supplementary Fig. S7), we observed that Ti$_3$C$_2$T$_x$ MXene made using graphite as the carbon source in the MAX phase (unlike previous work[37]) showed only partial etching for 5% HF, while 9.1% HF and 12.5% HF showed complete etching for the same temperature and reaction time (35 °C for 24 h), likely due to the higher HF concentrations[38]. Overall, previous work shows that the etching of MAX phases to MXenes begins at the exterior surface of the grain[39], which would mean that the exterior flakes of MXene are exposed to the acid mixture for a longer duration than the interior flakes—this may serve to increase the number of vacancies in the exterior flakes. If this hypothesis is true, it suggests that the delaminated 5% HF Ti$_3$C$_2$T$_x$ MXene flakes were delaminated from these etched MXene sheets nearer to the surface of

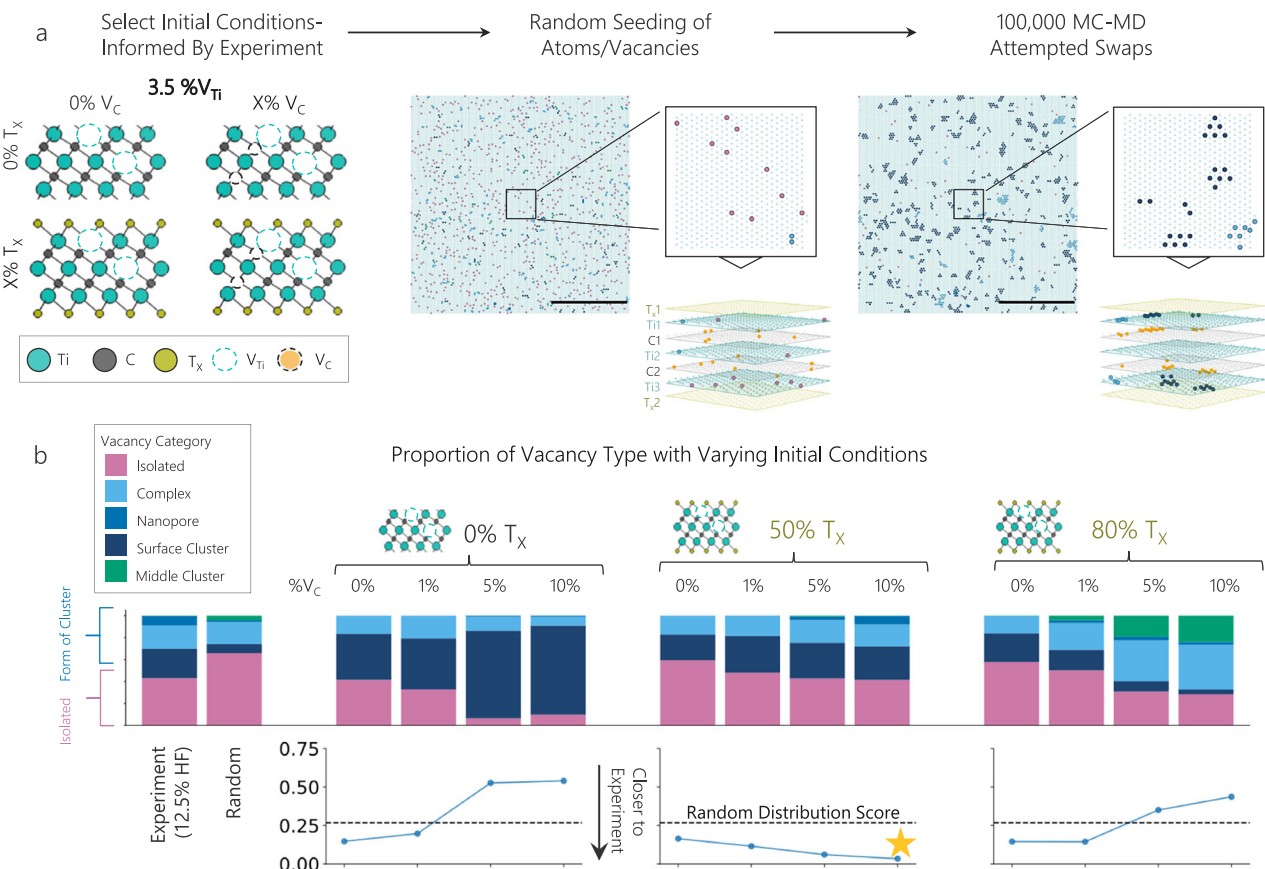

**Fig. 3 | Assessing vacancy behavior dynamics. a** Workflow for Monte Carlo (MC)–Molecular Dynamics (MD) simulations. A $V_{Ti}$ concentration of 3.5% (from 12.5% HF experiments) was used. Candidate $Ti_3C_2T_x$ grids were seeded with varying $V_C$ concentrations (0, 1, 5, 10%) and surface termination ($T_x$) levels (0, 50, 80%). Example shown in random and final MC step configurations (scale bars: 10 nm). Bolded dots represent vacancies, color indicates vacancy category; C vacancies (orange) and Ti vacancy categories, including isolated (pink), complex (light blue), nanopore (medium blue), surface cluster (dark blue), and middle cluster (green). **b** Vacancy clustering in relaxed configurations compared with experiment and random. The Y axis on line plots shows the Total Variation Distance (TVD) between clustering distributions from the 12.5% HF sample and corresponding MC–MD simulations shown directly above each point, with the random TVD score (dotted line) as reference. Y axis is shared across the line plots. Yellow star indicates run with the best TVD score (50% $T_x$, 10% $V_C$).

the grain. While beyond the scope of this study, these results indicate that etching duration, in addition to etchant concentration, is a key parameter for controlling 3D defect topology, warranting future investigation. Regardless, future studies investigating the role of flake locality and defect formation mechanisms based on the MAX grains are needed to better test this potential explanation.

This in-depth analysis of MXene vacancies underscores a critical need: leveraging AI-guided electron microscopy to map defects across all constituent atomic planes. This capability is essential for systematically quantifying and classifying defect populations and directly correlating them with specific synthesis conditions. Our analysis shows that clustered defects constitute nearly half of all present defects in all MXenes synthesized in this work, regardless of HF content (Fig. 2b), which can have major implications on our fundamental understanding of MXenes' electrical, thermal, and electrochemical properties. To advance our understanding of this vacancy behavior in MXenes, we next used computational methods to investigate why clustering is the preferred vacancy type in 2D MXenes.

**Energetic drivers of vacancy clustering**
Material entropy predicts the presence of defects, but the harsh processing conditions drive the observed quantities of vacancies. The distribution of these vacancies is driven by the energy environment around the vacancies. Recent power laws for the elastic field of vacancies in MXenes describe this energy environment[40]. Extending this environment allows us to isolate the key energetic drivers of

vacancy formation and gain understanding of the energetic stability of defect clustering. We used a hybrid Monte Carlo (MC) Molecular Dynamics (MD) method directly informed by our experimental observations. This approach allowed us to probe the influence of factors not directly visible in HAADF-STEM imaging, namely carbon vacancies ($V_C$) and surface terminations ($T_x$), on the experimentally observed Ti vacancy ($V_{Ti}$) distributions, as shown in Fig. 3a.

When selecting initial conditions for our MC-MD runs, we chose to seed 3.5% Ti vacancies ($V_{Ti}$), replicating what we saw experimentally in the 12.5% HF sample. Runs with 1.5% Ti vacancies, replicating the 5 and 9.1% HF $V_{Ti}$ concentrations, are included in Supplementary Fig. S8. We chose to vary the percent carbon vacancies ($V_C$) and percent occupied surface terminations ($T_x$) to understand how these variables change the resulting relaxed vacancy distributions, selecting from 0, 1, 5, or 10% $V_C$ and 0, 50, or 80 % $T_x$. Next, we randomly seeded a 30 × 30 nm $Ti_3C_2T_x$ grid with our selected initial conditions. Finally, we ran the MC-MD for 100,000 attempted swaps, at which point the energy per atom curve flattens out, as shown in Supplementary Fig. S9. We recorded the final distribution of vacancy types for each run.

In Fig. 3b, we visualize the differences between the vacancy distributions of our experimental data, the random 3.5% $V_{Ti}$ distribution, and the 12 different MC-MD runs. We calculate a total variation depth (TVD) similarity score for each of the runs as well as the random distribution to our experimental data. We find that the number of clustered Ti vacancies increases as the percent of carbon vacancies increases, across all surface termination percents (0, 50, 80%). This

trend is energetically favorable, as the co-location of carbon and titanium vacancies minimizes the number of broken bonds in the lattice. However, when comparing runs with equal carbon vacancy concentration, the number of clustered Ti vacancies decreases with additional surface terminations, and the clusters shift from the surface to the middle layer or inter-layer orientations. This behavior is likely due to the increase in the number of bonds connected to the outer Ti layers (M') as the surface termination concentration increases, making it less favorable for vacancies to congregate in the outer layer.

Overall, the 3.5% $V_{Ti}$ run closest to our 12.5% experimental results was the 50% occupied surface terminations,10% carbon vacancy run (TVD score of 0.09, compared to random TVD score of 0.29), indicating the importance of $T_x$ and $V_C$ vacancies in facilitating clustering of $V_{Ti}$ vacancies. Similarly, we found the 1.5% $V_{Ti}$ run closest to our 5 and 9.1% HF experimental results to be the 50% occupied surface terminations, 5% carbon vacancy run (TVD score of 0.06, compared to random TVD score of 0.31, shown in Supplementary Fig. S8), highlighting the positive correlation between of Ti and C vacancies as the HF etchant increases. Applying our previously defined 3D vacancy motifs to a theoretical framework allowed us to gain insight into the aspects of the MXene that we were unable to experimentally visualize, and more deeply understand the mechanisms that drive defect clustering in 2D materials.

## Discussion

Despite their importance in determining the properties and behavior of multi-layered 2D materials, such as MXenes, 3D understanding of point defects is a fundamental challenge in materials science. Here, we have established a framework where AI-guided microscopy reveals the 3D topology of vacancies and enables their statistical classification, moving far beyond the limited scope and potential bias of traditional manual analysis. Moreover, this approach is potentially suitable for even low-dose imaging, which can preserve the integrity of the underlying lattice and allow us to directly interrogate the intrinsic material. Using this method, we reveal unique trends in the MXene point defect population that result from processing, including changes in defect density and clustering that are related to changes in surface termination of the crystal. This large-scale experimental data provides critical ground truth for constraining and validating theoretical models, as demonstrated by our MD simulations, which elucidated the energetic drivers of clustering.

The deconvolution of the layers of Ti was made possible due to the three-layer structure of $Ti_3C_2T_X$ and the ability to position the beam to simultaneously visualize all three of these layers. Extending this method to other two-dimensional materials would require case-by-case modifications, particularly for materials that possess a different number of layers and lack comparable imaging conditions. Nevertheless, this study indicates that similar strategies may be applicable to a wide class of multi-layered 2D materials. Furthermore, it may be possible to combine this method with confocal imaging, focusing the imaging probe on different layers of the sample and thereby allowing us to reconstruct thicker crystals[41]. The extraction of 3D defect topology and clustering behavior can provide deeper insights into the formation and effects on material properties in these systems. This approach also represents a platform upon which to investigate the effects of atomic defects on the surrounding lattice, informing advanced defect processing strategies in emerging autonomous platforms. Ultimately, the integration of theory and experiment, facilitated by statistics and ML, yields a richer understanding of point defect clustering and dynamics, informing higher fidelity modeling and prediction of material performance. This provides a clear pathway toward the rational design of defect topologies to optimize performance in applications ranging from catalysis and energy storage to biomedicine.

## Method

### Synthesis

To synthesize the MXenes used in this study, we first synthesized a graphite-based Ti3AlC2 MAX phase precursor using molar ratios of 3:2:2 of Ti:Al:C, by first ball milling elemental Ti (325 mesh, Alfa Aesar), aluminum (325 mesh, Alfa Aesar), and graphite at a 2:1 ball-to-powder mass ratio using yttria-stabilized zirconia grinding balls in a high-density polyethylene (HDPE) container. After ball milling this graphite-based Ti3AlC2 MAX phase, we followed the typical synthesis protocol for Ti3C2Tx MXene, as discussed in-depth in a previous step-by-step article from our group[37]. In short, this mixed powder was placed into an alumina crucible inside an alumina tube furnace and sintered at 1400 °C for 4 h under 100 mL/min Ar flow. After sintering, the MAX block was drilled into a fine powder and washed using 3 M hydrochloric acid (HCl) at a volumetric ratio of 30 mL/g MAX powder overnight to remove any intermetallic impurities. After HCl washing, the powder was washed to a neutral pH using repeated centrifugation using deionized (DI) water, filtered, dried in air, and sieved using a 71 mm pore size sieve. After sieving, the powder was then placed into a solution containing 5% HF (3 mL HF/g MAX), 9.1% HF (6 mL/g MAX), or 12.5% HF (9 mL/g MAX) with 18 mL 12 M HCl/g MAX and 9 mL DI water per gram MAX and stirred at 35 °C for 24 h. After 24 h, the solution was centrifuged to neutralize the pH of the solution and placed into an aqueous solution containing 1 g of LiCl per gram starting MAX to 50 mL of DI water and stirred overnight at room temperature to delaminate the MXene into single flakes. After delamination, the solution was washed again using centrifugation to remove the excess Li, vortex mixed for 30 min, and finally run at 2380 RCF acceleration for 30 min to yield the single-to-few flake solution of Ti3C2Tx MXenes. These MXenes were then used to prepare for STEM imaging. To quickly screen the quality of these MXenes, X-ray diffraction (XRD) was used (Anton Paar XRDynamic 500, monochromatic Cu Kα radiation, 3–65° 2θ full range, 25 s/step).

### STEM imaging

STEM samples were prepared by diluting MXene solutions and drop casting onto ultrathin carbon on lacey carbon support TEM grids. High-angle annular dark-field (STEM-HAADF) images of the samples were acquired using a probe-corrected Thermo Fisher Scientific Spectra 200 STEM operating at 60 kV with a convergence semi-angle of 30 mrad. The beam current was limited to 10–25 pA to minimize sample damage. STEM-HAADF images were acquired as stacks of low dwell time frames, which were subsequently rigidly aligned to obtain high signal-to-noise ratio (SNR) images. Rigid image registration was performed using a method optimized for very low SNR images, like those collected here, to minimize the possibility of unit cell jump errors, which would inhibit vacancy analysis[42].

To minimize the buildup of carbon contamination during high-resolution imaging, the prepared grids were first cleaned with acetone and methanol and then dried under vacuum for at least 12 h before loading into the STEM. During imaging, beam showering in TEM mode was used to further prevent contamination buildup. Atomic-resolution imaging was not possible before beam showering, making a before-and-after comparison impossible. However, during imaging we often repeated the beam shower step multiple times over a similar field of view and did not observe any noticeable changes between repetitions.

For each sample, the grid was surveyed upon loading in the TEM to check for any anomalous areas. Subsequently, one to two monolayer flakes were randomly chosen for atomic-resolution data collection and defect analysis. All images are shown in Supplementary Figs. S10–12. We note that many of the images exhibit some "striping" in the MXene lattice. This is due to rippling in the MXene flakes causing slight local tilt of the c axis. The same effect was previously demonstrated by Sang et al.[30].

## Machine learning and statistical analyses

We used AtomAI's Segmentor class (atomai.models.Segmentor) as our framework, which defines a U-Net neural network for semantic segmentation of microscopy images[20]. AtomAI, a Python package built on top of PyTorch for deep learning in microscopy, is typically used to locate the positions of atoms in STEM images and has been applied to identify impurity atoms and defect locations in graphene[20]. Due to the beam sensitivity and surface contamination of our MXene samples, it was more beneficial to train a network to pick out defects than atoms; therefore, we approached the training of our models differently than previous graphene work. We trained two models (1) Lattice Capture Neural Network and (2) Defect Spotter Neural Network. We used the first model to locate the positions of atoms if our image was pristine, hexagonal lattice and we used the second model to label vacancies. This two-model approach proved more robust against variations in image contrast and local disorder than a single model trained to identify both atoms and defects simultaneously.

A major challenge of training models in materials science is the lack of labeled training data[35,43,44]. Specifically for this study, quality STEM images of MXenes are difficult and time consuming to take. Therefore, we did not have access to a large training dataset, regardless of whether it was labeled or unlabeled. Our own data was time-consuming to image, and we did not want to lose valuable information that could be utilized in our statistical analyses. In addition, we wanted to create a model that was robust to change across experimental settings, so it could be reused across a variety of MXene images. Each STEM instrument has its own environmental effects, and these are subject to change over time. Therefore, we'd like to create a model that works generally for MXene images and is not specialized for one instrument's data on a particular day. To accomplish this, we turned to a previous paper[30], and used crops from one high resolution, large frame of view image to train our models. We then used 2D Gaussian fitting to label the data (note we were only able to do this because of the sharpness of the large image and 2D Gaussian fitting was unreliable for our own data), giving us binary masks of atomic positions in the cropped images. The crops were square with length randomly chosen between 150 and 300 pixels, allowing the model to be applicable across a range of resolutions. Crops were then resized to $256 \times 256$ pixels ($2^n$ being an ideal size for a neural network). Data augmentation, including rotation and Gaussian noise, was incorporated as part of the neural network's training process. This step was imperative to the success of the model outside of the training dataset. It was necessary for the model to be successful on MXene images taken on a completely different STEM instrument, allowing for a range of lattice directions, resolutions and sharpness.

The Lattice Capture Neural Network was trained as 3 ensembled models each with 1000 training cycles of batch size 15, trained on 1000 crops of our parent image. Gaussian noise in range [40,60] and rotation were incorporated in the data augmentation. The Defect Spotter Neural Network was one model, trained with 350 training cycles and 300 crops of our parent image. Gaussian noise in range [40,100], rotation, and jitter were incorporated in the data augmentation. All training data and model weights are included in the files and code to run the models is included in Jupyter Notebooks in our GitHub repository.

It is also important to note is that our ML models were trained to locate all atomic positions, including the Ti adatoms that can be seen in our images. The "brightness" of the adatoms can cause the surrounding atoms to look washed out, which can influence the performance of the model if it is not properly trained. To address this, we included a higher proportion of crops from our source image that contained adatoms, so our training dataset would have enough representation of the environment that surrounds the bright adatoms and learn to correctly identify these atomic positions. For the purposes of this paper, we focused on identifying and describing the vacancies;

however, future work could investigate the relationship between vacancy and adatom positions.

Once we obtained all atomic/defect locations, we (1) differentiated the middle (M″) from outer (M′) layers and (2) combined this with a Delaunay triangulation to define 3D defect motifs. For (1), we enforced a hexagonal structure in our dots using their lattice directions (when finding the dots using ML, the structure was not perfectly hexagonal and sometimes morphed a little; future work could investigate the mechanisms causing this). Then, we separated each dot into its relative layer (i.e., we noted that dot 1 and dot 2 must be in the same layer due to the projection pattern of the layers, but we could not yet say which layer that was). For each image, we now had three groups of relative layers, and we counted the number of defects in each of these. We consistently saw one layer with significantly less defects than the other two and were able to label this layer as the middle layer (M″), since previous theoretical studies[36] have shown defects are more likely in outer layers. The other two layers were both classified as outer layers (M′). We could not differentiate between the upper or lower layers; instead, we categorized the one with more defects as M′$_{max}$ and the one with less as M′$_{min}$. For (2), we took these layer classifications and conducted Delaunay triangulations on each layer, as well as the projection of all three. This allowed us to understand the connectivity between defects within layers as well as between layers, which led to the motifs we defined in Fig. 2 of the main text.

## Modeling

Using the Large-Scale Atomic/Molecular Massively Parallel Simulator (LAMMPS)[45], we conducted hybrid Monte Carlo (MC) Molecular dynamics (MD) calculations. The bond-order potential (BOP) developed by Plummer et al.[46,47]. simulated all interatomic interactions. We started with a large (30nmx30nm) supercell of pristine MXene ($Ti_3C_2$) to minimize edge effects and establish statistical significance. The nanosheet was seeded with a random distribution of Ti vacancies, C vacancies, and surface terminations (O and F). Cases were populated with 1.5% and 3.5% Ti vacancies to recreate the experimentally observed conditions. Since carbon is difficult to resolve experimentally, C vacancies were seeded with 0, 1, 5, and 10%. This is in line with experimental suggestion[48] and extends to higher concentrations observed in aged MXenes[26]. Surface termination distributions were also randomly seeded. Preliminary calculations showed that the different species (O, F, and vacancy) orient randomly. Additionally, the O and F termination bonds in the interatomic potential are similar enough that there is not a significant energy difference, so we simplified to a single species (O). We chose three surface termination coverages 0, 50, and 80% in order to be in line with literature[49,50] and examine the impacts of varying levels of coverage on Ti vacancy clustering. The MC model does not explicitly prevent surface terminations from occupying Ti vacancy sites. However, the energetic relaxation ensures that there are no isolated surface terminations.

The vacancies are represented by non-interacting ghost atoms. These atoms are placeholders in the lattice that have no velocity or forces to contribute to the energy of the system. Placeholders are needed to perform Monte Carlo (MC) swaps to rapidly sample and compare energy configurations. Previous works have used MC methods to examine point defects[51], and some MD codes use ghost atoms, but their use to track vacancy configurations is a nuance of our approach. Each MC calculation included 100,000 attempted swaps. Every step, one candidate atom is randomly selected from each eligible group. The candidate groups are paired and include Ti and ghost-$V_{Ti}$, C and ghost-$V_C$, O and ghost-$V_T$ (surface termination). For each grouped pair of atoms, their positions are reversed and the potential energy calculated according to the interatomic potential. Simply, the potential describes the energy between any two atoms at a given distance under the influence of an outside atom. The sum of these energies is the system potential energy. After the swap, if the potential energy is

lower, the swap is accepted. If it is not, then there is a chance of acceptance based on the Metropolis criterion at 300 K, which is summarized by Eq. 1.

$$R < e^{\frac{-\Delta E_{swap}}{k_B T}} \qquad (1)$$

Where R is a randomly generated number between 0 and 1, $\Delta E_{swap}$ is the change in energy from the swap, $k_B$ is the Boltzmann constant, and $T$ is the temperature[52]. If it is still not accepted, then the two candidates are returned to their starting location, and another pair is tested. After an accepted swap, that is the new starting configuration for the next step. In this way, the distribution progression follows a path of swaps governed by the random seed candidate atom selection and entropic acceptance.

Cases included 100,000 attempted swaps because convergence the potential energy converged at this point for the 30 × 30 nm samples. Fig. S9 shows this convergence. The number of swaps needed to show convergence is directly proportional to the number of candidate atoms. Additional studies were conducted with fewer atoms and relatively more swaps; these showed similar potential energy behavior but were not as statistically robust.

## Data availability
The raw STEM images, trained machine learning models, 3D animations of defect visualizations, and molecular dynamics simulation input and output files data generated in this study have been deposited in the Figshare database (https://doi.org/10.6084/m9.figshare.30385891). The processed data generated in this study are provided in the Supplementary Information/Source Data file. Source data are provided with this paper.

## Code availability
The associated codebase[53] is available on GitHub and can be referenced through Zenodo under https://doi.org/10.5281/zenodo.18865181.

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

## Acknowledgements

G.G. would like to thank Dirk Jordan for useful discussions regarding the bootstrapping analysis. G.J.T. is grateful for the additional support of this work by the Eula Mae and John Baugh Chair in Physics at Baylor University. This work was authored in part by the National Laboratory of the Rockies (NLR) for the U.S. Department of Energy (DOE), operated under Contract No. DE-AC36-08GO28308. G.G., M.S., A.G., H.E., and S.R.S. were supported by the Operando to Operation (O2O) Laboratory Directed Research and Development (LDRD) program at NLR. G.G. was supported by the U.S. Department of Energy, Office of Science, Office of Workforce Development for Teachers and Scientists (WDTS) under the Science Undergraduate Laboratory Internship (SULI) program. B.C.W. is supported by the Laboratory Directed Research and Development (LDRD) of Argonne National Laboratory, Office of Science, US Department of Energy (contract 2026-0334). S.R.G. is supported by the Department of Defense (DoD) through the National Defense Science and Engineering Graduate (NDSEG) Fellowship Program. B.C.W and B.A. acknowledge funding support from the U.S. NSF, award number CMMI-2134607. The views expressed in the presentation do not necessarily represent the views of the DOE or the U.S. Government. The U.S. Government retains, and the publisher, by accepting the article for publication, acknowledges that the U.S. Government retains a nonexclusive, paid-up, irrevocable, worldwide license to publish or reproduce the published form of this work, or allow others to do so, for U.S. Government purposes.

## Author contributions

G.G. and S.R.S. conceived the project. B.C.W. and B.A. led the synthesis of the MXene materials. M.A.S. performed the electron microscopy experiments. S.G. and G.J.T. led the molecular dynamics simulations. G.G. developed the machine learning methodology, with input from H.E. and A.G, analyzed the data, and wrote the original manuscript. S.R.S. supervised the project. All authors discussed the results and contributed to the revision of the manuscript.

## Competing interests

The authors declare no competing interests.
