## [Transparent Peer Review file · Nature Communications]

Revealing the Hidden Third Dimension of Point Defects in Two-Dimensional MXenes

Corresponding Author: Professor Steven Spurgeon

Version 0:

Reviewer comments:

Reviewer #1

(Remarks to the Author)

The manuscript draft presents an AI-guided STEM workflow to reconstruct the 3D topology of point defects in $\text{Ti}_3\text{C}_2\text{T}_x$ MXene, addressing a long-standing challenge in the characterization of multilayer 2D materials. The authors combine large-scale vacancy mapping with statistical analysis and molecular dynamics simulations to reveal point defect structures and their influence on the sample synthesis conditions. Overall, the study is timely and of broad interest to the materials characterization and 2D materials communities. The integration of artificial intelligence with STEM represents a valuable methodological advance. With further clarification, this work has strong potential to make a significant impact in the field.

1. The authors focus primarily on vacancy analysis, however, the experimental STEM images clearly contain numerous high-contrast features that appear to correspond to Ti adatoms. These “point” defects are not discussed or analysed, and their presence should at least be acknowledged. An important question that arises is whether the occurrence of adatoms correlates with the presence of vacancies. Additionally, it would be helpful for the authors to clarify how their workflow interprets the intensities associated with Ti adatoms and whether these features influence the accuracy of vacancy identification in their vicinity.
2. The authors claim to resolve the 3D defect distribution in a multilayer 2D material. However, after reading the manuscript, it is not fully clear how accurate this statement is. For example, in the section on “classification of 3D defect motifs” for isolated vacancies, can the method reliably distinguish whether a vacancy resides in the electron-entrance surface layer, a middle layer, or the electron-exit surface layer? Is it important to state how multilayer 2D material is positioned with respect to the electron beam? By having just one type of defect (hypothetical example, only isolated Ti vacancies just in one layer) such analysis would not be able to distinct in which layer unless other types of ‘cluster types’ are present, same applies for surface cluster and inter-layer. Can authors elaborate on this?
3. Authors claim that certain number of images is statistically meaningful to get the representative result. However, MXene could be rather inhomogeneous in features and could depend where analysis is done on the TEM grid. Are the images presented in the manuscript where acquired from one isolated area on the grid? Or randomly chosen (far apart separated) positions on the TEM grid? How and if this should affect the analysis results? Can beam shower, which authors used, to induce the additional point defects? Please provide those details to the manuscript draft.

(Remarks on code availability)

Reviewer #2

(Remarks to the Author)

Review:

This manuscript sets out to image and resolve the 3-dimensional arrangement of defects in 2D materials using AI-guided electron microscopy. The authors make their argument using $\text{Ti}_3\text{C}_2\text{T}_x$ MXene as a prototypical 2D material. They argue that MXenes in general would greatly benefit from this advancement due to the diverse properties observed in MXenes and the wide applications spaces developing for MXenes. They synthesized several $\text{Ti}_3\text{C}_2\text{T}_x$ MXene samples with deliberately different point defect concentrations and performed STEM imaging to build sets of atomic resolution images for each

specimen. Afterwards they used a neural network for semantic segmentation in order to identify the point defects in each acquired STEM image. Their NN method was successfully able to identify many thousands of atoms and point defects across their STEM images and also, based on the real space positions of the Ti-atomic columns, identify whether the point defect was on a Mxene surface layer or in the middle layer. Next, they classified the 3-dimensional vacancy arrangements in order to study the vacancy clustering and observed 4 specific vacancy motifs. They then use computation methods to investigate why vacancy clustering is the preferred vacancy motif observed using Monte Carlo Molecular Dynamics. The MCMD simulations find that C vacancies behave similarly to Ti vacancies, but the effect of Tx surface terminations is quite striking and the more Tx surface terminations present cause fewer surface clustered Ti vacancies. This reviewer believes the work presented here is interesting and highly relevant to the readership of Nature Communications. Obtaining 3-dimensional information from atomic resolution imaging techniques is a long-standing goal of the microscopy community and progress to that end is worthy of consideration and will attract wide readership. While I have a few comments recorded below, I recommend this manuscript for publication with minimal revision.

My individual comments are below:

-In the SI you mention using a beam shower in TEM mode prior to STEM analysis of the prepared Mxene sheets. Do you have a sense of whether or not the beam shower, which is typically just leaving a spread parallel beam on an area for an amount of time, induced any changes in your sample? In my experiences analyzing MXenes I have noticed that the electron beam can impact and alter the Tx on the flake surfaces, some good examples in literature are publications by Xie (J. Am. Chem. Soc. 136, 6385–6394 (2014)) and Hart (Nat. Commun. 10(1), 522 (2019)). Did you have success looking at areas that were not subject to a beam shower or was the contamination too great?

-It is very clever to deconvolve the constituent Ti layers in HAADF images into M' and M'' layers. It makes sense that the inner most layer would be the one with the least point defects and can be identified as such. The following sentence is confusingly worded:

"This analysis shows an average increase of M' and M'' vacancies from 2.05 at% to 0.21 at% in 5 wt% HF, respectively, from 5.15 at% to 0.69 at% in the 12.5 wt% HF samples, respectively (Fig. 1e)."

Use of the word "to", like when saying "from 2.05 at% to 0.21 at%" seems misplaced and could more clearly be read if changed to "and" in the current form it appears to imply that the number of vacancies have a range or are decreasing, rather than there are a specific amount of M' and M'' vacancies present.

-Regarding the statement "However, we further hypothesized that if over-etching were to occur due to a higher concentration of HF in the etchant, as suggested in previous works^{26,28}, it would result in vacancy clustering rather than isolated point vacancies, as atoms around vacancy sites would likely be easier to remove than those with a pristine surface surrounding them."

In the context of the rest of the 2D/TMD atomic resolution STEM literature, what concentration of point defects do you think would be present in even the best Mxene flakes? Most single flakes of non-single element 2D materials have tons of vacancies, and high-quality STEM work typically causes more of them to form. Specialized synthesis routines can create relatively point-defect free TMDs but within the size of the STEM images shown, there's usually at least 1 point defect. I would expect that most typical Mxenes are similar. I understand you took the best steps possible to ensure that the STEM probe isn't inducing point defects and agree that you did as much as possible to rule that out. While this is likely beyond your paper but what would a "best" synthesis of a Mxene flake look like with regard to your 4 defect motifs description of point defects in Mxene? A baseline of some kind of population of point defects from which to say is thermodynamically/kinetically present would help ground a reader to understand the quality of the flakes and the reality of point defects in thin 2D materials.

The MD simulations add a lot here, I understand your setup for the simulations is based on the STEM imaging, but is there any theoretical observations or calculations that suggest that the observed point defect concentration from STEM are the expected distributions of point defects from a typical exfoliated flake? They likely highly depend, as you mention, on the processing conditions (which are numerous!) but I feel that the MD simulations shown here offer some clearer guidance for readers in the discussion.

-Regarding the statement "Overall, previous work shows that the etching of MAX phases to MXenes begins at the exterior surface of the grain³⁶, which would mean that the exterior flakes of MXene are exposed to the acid mixture for a longer duration than the interior flakes—this may serve to increase the number of vacancies in the exterior flakes. If this hypothesis is true, we believe this could suggest that the delaminated 5% HF Ti₃C₂T_x MXene flakes were delaminated from these etched MXene sheets nearer to the surface of the grain. While beyond the scope of this study, these results suggest that etching duration, in addition to etchant concentration, is a critical parameter for controlling 3D defect topology, warranting future investigation." While I follow your logic and can accept the argument - that 5% HF samples are likely flakes closer to the exterior of the parent MAX grain (because the delamination was not complete) so they could/should have more vacancies, it is not clear what is exactly meant by this statement as the data shown in Fig. 2 seems to show that the 5% HF sample had the fewest point defects of the 3 samples. Can you clarify what is meant here?

(Remarks on code availability)

I attempted to visit the code at https://github.com/nrel/mxene_seg and unfortunately the link was down and I received a 404 error. Perhaps the link is non-public while the manuscript is under consideration?

Reviewer #3

(Remarks to the Author)

The authors investigate vacancy concentration and distribution in MXenes by machine-learning guided analysis of scanning tunneling electron microscopy (STEM) images, complemented by vacancy clustering simulations.

While the main findings of the analysis appear sensible, I get the impression that lot of the potential problems are swept under the rug. This is especially true for the first section of Results.

About STEM imaging:

- It's not clear to me what the "stacked STEM images" are? I first thought it means that images are taken from the same location, but the stacked images in Supp. Figures 8-10 look very different.
- Actually, it's also never clearly stated that the imaged flakes are monolayers, although I presume that to be the case.
- Do the authors track whether vacancies are formed during imaging?
- Are the images taken from flake interior or exterior, and does the defect concentration depend on the location?
- Is it possible to corroborate the vacancy or termination concentration trends via e.g. EDX/XPS measurements?
- No explanation is given to the obvious "striping" in all STEM images. Could it be related to the functional group distribution?

About ML model:

- Little details are given in the main text about how the vacancy layers is determined. The authors can identify vacancies in 3 different layers, since Ti atom in each layer has different lateral position, but they cannot say which layer is which. The low concentration layer is always taken to be the middle layer. This may very well be correct, but it's nevertheless an assumption that is important to lay out in the main text. Consequently, claiming that a map of 3D topology is created is a bit questionable.
- If I understood correctly, the ML model was trained using different crops from one microscopy image from an old paper. This sounds dubious, both for the small amount training data and the fact that all the experimental details are slightly different. Since the authors have a lot of own images, why not use those?

About the MC simulations:

- The authors use bond-order potential developed for MAX phase. Here it is applied to MXenes without any testing on its accuracy, in particular how it works for the terminations.
- The authors do not mention if there is a correlation between the Ti vacancy formation with the terminations in the neighboring sites?

All that said, I believe the extracted vacancy positions are still largely valid and their clustering can then obviously be analyzed. This part of the manuscript provides clear trends on the types of vacancies/vacancy clusters and their dependence on the HF concentration, which should be of considerable interest to the community.

However, since there are many technical details missing and the validity of the approaches insufficiently substantiated, I think the manuscript is not suitable for Nature Communications in the present form.

(Remarks on code availability)

I could not access the site.

Version 1:

Reviewer comments:

Reviewer #1

(Remarks to the Author)

I thank manuscript authors for addressing my comments and revising manuscript accordingly. With these changes, I recommend the manuscript for publication in its current form.

(Remarks on code availability)

Reviewer #2

(Remarks to the Author)

I thank the authors for responding to the comments of myself and the other reviewers. After considering their responses to our comments, I can easily recommend this manuscript for publication in Nature Communications.

(Remarks on code availability)

This time I was able to view the source code and from a quick review of the repo page it is clear to me that I could easily clone the repository and engage with the code in a local environment.

Reviewer #3

(Remarks to the Author)

The authors have revised the manuscript in response to comments and question from three Reviewers.

In my previous report, I concluded that the vacancy clustering analysis is likely valid and useful to the community, but due to missing experimental details and problems in the computational methodology, I thought the manuscript was not suitable for Nature Communications.

In the revised version, the authors have provided additional details and clarifications concerning the experimental part.

One of my main criticisms was that the bond-order potential developed for MAX phases was used these MXene systems without testing. In fact, the potential was also valid for MXenes. It was just incorrectly described in the text and missing relevant reference.

The small amount of data used to train the ML model remains a deficiency, but overall I think the manuscript has improved enough to be suitable for publication in Nature Communications.

(Remarks on code availability)

January 14, 2026

Reviewer Response: NCOMMS-25-85357-T: “Revealing the Hidden Third Dimension of Point Defects in Two-Dimensional MXenes.”

Reviewer 1

The manuscript draft presents an AI-guided STEM workflow to reconstruct the 3D topology of point defects in $Ti_3C_2T_x$ MXene, addressing a long-standing challenge in the characterization of multilayer 2D materials. The authors combine large-scale vacancy mapping with statistical analysis and molecular dynamics simulations to reveal point defect structures and their influence on the sample synthesis conditions. Overall, the study is timely and of broad interest to the materials characterization and 2D materials communities. The integration of artificial intelligence with STEM represents a valuable methodological advance. With further clarification, this work has strong potential to make a significant impact in the field.

- 1. The authors focus primarily on vacancy analysis, however, the experimental STEM images clearly contain numerous high-contrast features that appear to correspond to Ti adatoms. These “point” defects are not discussed or analysed, and their presence should at least be acknowledged. An important question that arises is whether the occurrence of adatoms correlates with the presence of vacancies. Additionally, it would be helpful for the authors to clarify how their workflow interprets the intensities associated with Ti adatoms and whether these features influence the accuracy of vacancy identification in their vicinity.*

Our Response: We thank the Reviewer for pointing out the presence of the Ti adatoms in our STEM images and have added the following comment on them in section C of Supplementary Note 2.

Additional Text: “It is also important to note is that our ML models were trained to locate all atomic positions, including the Ti adatoms that can be seen in our images. The “brightness” of the adatoms can cause the surrounding atoms to look washed out, which can influence the performance of the model if it is not properly trained. To address this, we included a higher proportion of crops from our source image that contained adatoms, so our training dataset would have enough representation of the environment that surrounds the bright adatoms and learn to correctly identify these atomic positions. For the purposes of this paper, we focused on identifying and describing the vacancies, however, future work could investigate the relationship between vacancy and adatom positions.”

- 2. The authors claim to resolve the 3D defect distribution in a multilayer 2D material. However, after reading the manuscript, it is not fully clear how accurate this statement is. For example, in the section on “classification of 3D defect motifs” for isolated vacancies, can the method reliably distinguish whether a vacancy resides in the electron-entrance surface layer, a middle layer, or the electron-exit surface layer? Is it important to state how multilayer 2D material is positioned with respect to the electron beam? By having just one type of defect (hypothetical*

example, only isolated Ti vacancies just in one layer) such analysis would not be able to distinct in which layer unless other types of 'cluster types' are present, same applies for surface cluster and inter-layer. Can authors elaborate on this?

Our Response: We thank the Reviewer for allowing us to clarify some interesting points about our study. Using the hexagonal geometry of the MXene structure and the way it is positioned with respect to the beam (such that Ti atoms from all 3 layers are seen flattened into 2 dimensions) we could determine the respective layers of each atom. In other words, just from the structure of the material, we can say if two atoms are in the same layer. We then classify the layer with the least defects as the middle layer. This assumption is backed up by theory and our own data; we consistently see one layer with far fewer defects than the other two layers. However, we could not decipher the electron entrance surface layer from the electron exist surface layer.

For the hypothetical example given here, we would classify the isolated Ti vacancies in one layer as coming from one of the two outer layers. Cluster types reinforce our identification of the middle layer, because we see that many of the middle vacancies come from larger vacancies such as nanopores (and not from isolated middle clusters), however, the types of clusters were not considered when identifying the middle and outer layers.

There are some specific properties of MXenes that make it possible to apply these analyses, specifically the three layer structure and the ability to capture Ti atoms in all three layers in our images, as well as the number of vacancies (>1% of atomic positions). It is likely that the technique will need to be adapted to extend it to other materials that do not have these properties. Nevertheless, this study indicates that similar strategies may be applicable to a wide class of multi-layered 2D materials. Furthermore, it may be possible to combine this method with confocal imaging, focusing the imaging probe on different layers of the sample and thereby allowing us to reconstruct thicker crystals.

We agree that these points need to be elaborated on, and have added the following to the results section of main text:

Additional Text: “Given all these locations, we took advantage of the positioning of the material with respect to the beam (as seen in Fig. 2b) and the lattice angles associated with the hexagonal structure of the MXenes to determine the relative layer of each atom (further details on this tiling method in Supplementary Note 1 and Fig. S3). Next, we counted the number of vacancies in each relative layer and took the layer with the least number of vacancies to be the middle layer (M'') and the other two layers to be outer layers (M'). This assumption is backed up both by previous theoretical studies³⁶, which found that surface vacancies have a much lower formation energy than inner vacancies, as well as our own findings, which show one layer with far fewer vacancies than the other two layers, across all three HF samples (Fig. 1e).”

And the following to the conclusion section of the main text, as well as an additional citation (DOI: 10.1016/j.ultramic.2016.11.002):

Additional Text: “The deconvolution of the layers of Ti was made possible due to the three-layer structure of $Ti_3C_2T_x$ and the ability to position the beam to simultaneously visualize all three of these layers. Extending this method to other two-dimensional materials would require

case-by-case modifications, particularly for materials that possess a different number of layers and lack comparable imaging conditions. Nevertheless, this study indicates that similar strategies may be applicable to a wide class of multi-layered 2D materials. Furthermore, it may be possible to combine this method with confocal imaging, focusing the imaging probe on different layers of the sample and thereby allowing us to reconstruct thicker crystals⁴¹.”

3. *Authors claim that certain number of images is statistically meaningful to get the representative result. However, MXene could be rather inhomogeneous in features and could depend where analysis is done on the TEM grid. Are the images presented in the manuscript where acquired from one isolated area on the grid? Or randomly chosen (far apart separated) positions on the TEM grid? How and if this should affect the analysis results? Can beam shower, which authors used, to induce the additional point defects? Please provide those details to the manuscript draft.*

Our Response: TEM grids were prepared by drop casting MXene from well dispersed solutions, which were mixed just prior to drop casting. When loaded into the microscope, each grid was surveyed at low magnification to check for any anomalous areas. Subsequently, we randomly chose one to two monolayer flakes to collect atomic-resolution data from for defect analysis. Our analysis of the 9.1 wt% MXene sample includes images acquired from two grids on two different days. It did not reveal any observable differences between the samples. We take this as evidence that our imaging method for the other two samples was also representative. We have added the following statement to the STEM imaging methods section of the SI.

Additional Text: “For each sample, the grid was surveyed upon loading in the TEM to check for any anomalous areas. Subsequently, one to two monolayer flakes were randomly chosen for atomic-resolution data collection and defect analysis.”

We were unfortunately not able to collect atomic-resolution images of MXenes without beam showering, so we cannot make a direct before-and-after comparison. However, during imaging we often repeated the beam shower step multiple times over a similar field of view and did not observe any noticeable changes between repetitions. We did note, when testing different beam currents, that excessive beam current tended to cause amorphization of the film. We did not find any evidence of that amorphization due to the initial beam shower. We have added the following text to the STEM imaging methods section of the SI.

Additional Text: “Atomic-resolution imaging was not possible before beam showering, making a before-and-after comparison impossible. However, during imaging we often repeated the beam shower step multiple times over a similar field of view and did not observe any noticeable changes between repetitions.”

Reviewer 2

This manuscript sets out to image and resolve the 3-dimensional arrangement of defects in 2D materials using AI-guided electron microscopy. The authors make their argument using Ti₃C₂T_x Mxene as a prototypical 2D material. They argue that Mxenes in general would greatly benefit from this advancement due to the diverse properties observed in Mxenes and the wide applications spaces developing for Mxenes. They synthesized several Ti₃C₂T_x Mxene samples with deliberately different

point defect concentrations and performed STEM imaging to build sets of atomic resolution images for each specimen. Afterwards they used a neural network for semantic segmentation in order to identify the point defects in each acquired STEM image. Their NN method was successfully able to identify many thousands of atoms and point defects across their STEM images and also, based on the real space positions of the Ti-atomic columns, identify whether the point defect was on a Mxene surface layer or in the middle layer. Next, they classified the 3-dimensional vacancy arrangements in order to study the vacancy clustering and observed 4 specific vacancy motifs. They then use computation methods to investigate why vacancy clustering is the preferred vacancy motif observed using Monte Carlo Molecular Dynamics. The MCMD simulations find that C vacancies behave similarly to Ti vacancies, but the effect of Tx surface terminations is quite striking and the more Tx surface terminations present cause fewer surface clustered Ti vacancies. This Reviewer believes the work presented here is interesting and highly relevant to the readership of Nature Communications. Obtaining 3-dimensional information from atomic resolution imaging techniques is a long-standing goal of the microscopy community and progress to that end is worthy of consideration and will attract wide readership. While I have a few comments recorded below, I recommend this manuscript for publication with minimal revision.

1. *In the SI you mention using a beam shower in TEM mode prior to STEM analysis of the prepared Mxene sheets. Do you have a sense of whether or not the beam shower, which is typically just leaving a spread parallel beam on an area for an amount of time, induced any changes in your sample? In my experiences analyzing MXenes I have noticed that the electron beam can impact and alter the Tx on the flake surfaces, some good examples in literature are publications by Xie (J. Am. Chem. Soc. 136, 6385–6394 (2014)) and Hart (Nat. Commun. 10(1), 522 (2019)). Did you have success looking at areas that were not subject to a beam shower or was the contamination too great?*

Our Response: We thank the Reviewer for mentioning these references, which we have included as relevant prior work. Unfortunately, the contamination was too great before beam showering to acquire atomic resolution images. However, during imaging we often repeated the beam shower step multiple times over a similar field of view and did not observe any noticeable changes between repetitions. In the suggested publication, Hart, et al. report minimal sample damage from the parallel beam used for their EELS measurements, which should be very similar to the relatively low magnification, parallel beam used here for beam showering. They note that beam exposure appeared to transform -OH termination groups to the more stable =O but seem to find that alteration of =O and -F groups required high temperature (>500 C) annealing.

We did see some beam-induced damage during STEM imaging at higher currents than used to collect the images reported here. This damage tended to lead to amorphization of the MXene structure, which was not observed after beam showering nor at the 10-25 pA currents used here for STEM imaging. Thus, we are confident there were no significant effects on the MXenes due to the beam shower procedure.

2. *It is very clever to deconvolve the constituent Ti layers in HAADF images into M'' and M' layers. It makes sense that the inner most layer would be the one with the least point defects and can identified as such. The following sentence is confusingly worded: "This analysis shows an average increase of M' and M'' vacancies from 2.05 at% to 0.21 at% in 5 wt% HF, respectively, from 5.15 at% to 0.69 at% in the 12.5 wt% HF samples, respectively*

(Fig. 1e).”Use of the word “to”, like when saying “from 2.05 at% to 0.21 at%” seems misplaced and could more clearly be read if changed to “and” in the current form it appears to imply that the number of vacancies have a range or are decreasing, rather than there are a specific amount of M’ and M’’ vacancies present.

Our Response: We agree with the Reviewer that the sentence is confusingly worded and have changed the sentence as they suggested to, “ This analysis shows an average increase of M’ and M’’ vacancies from 2.05 at% and 0.21 at% in 5 wt% HF, respectively, to 5.15 at% and 0.69 at% in the 12.5 wt% HF samples, respectively (Fig. 1e).”

3. *Regarding the statement “However, we further hypothesized that if over-etching were to occur due to a higher concentration of HF in the etchant, as suggested in previous works 26,28, it would result in vacancy clustering rather than isolated point vacancies, as atoms around vacancy sites would likely be easier to remove than those with a pristine surface surrounding them.” In the context of the rest of the 2D/TMD atomic resolution STEM literature, what concentration of point defects do you think would be present in even the best Mxene flakes? Most single flakes of non-single element 2D materials have tons of vacancies, and high-quality STEM work typically causes more of them to form. Specialized synthesis routines can create relatively point-defect free TMDs but within the size of the STEM images shown, there’s usually at least 1 point defect. I would expect that most typical Mxenes are similar. I understand you took the best steps possible to ensure that the STEM probe isn’t inducing point defects and agree that you did as much as possible to rule that out. While this is likely beyond your paper but what would a “best” synthesis of a Mxene flake look like with regard to your 4 defect motifs description of point defects in Mxene? A baseline of some kind of population of point defects from which to say is thermodynamically/kinetically present would help ground a reader to understand the quality of the flakes and the reality of point defects in thin 2D materials.*

Our Response: We thank the Reviewer for their insightful points and opportunities to improve the quality and communication of this work. Thus far in the MXene community, our best measured quality flakes have observed a Ti defect concentration of < 0.9 at% via SIMS (DOI: 10.1038/s41467-024-50713-2) in high-quality “pristine” MXenes but can range as high as ~20 at% in highly defective and HF-damaged flakes. In part due to this article and other ongoing works from both our team and the entire community, the knowledge of MXenes’ defects is still growing, and we will understand this baseline much better in the next few years. However, this work is the first of its kind showing defects in all three atomic planes in the MXene materials, which is critical to both these future works in MXenes as well as other 2D materials.

To add this baseline discussion, we have added the following discussion in the main text:

Additional Text: “These values are on par with previous studies demonstrating a ~1 at% vacancies in the M’ layer for high-quality MXenes.²⁶”

4. *The MD simulations add a lot here, I understand your setup for the simulations is based on the STEM imaging, but is there any theoretical observations or calculations that suggest that the observed point defect concentration from STEM are the expected distributions of point defects from a typical exfoliated flake? They likely highly depend, as you mention, on the processing*

conditions (which are numerous!) but I feel that the MD simulations shown here offer some clearer guidance for readers in the discussion.

Our Response: We thank the Reviewer for giving us the opportunity to expand on this point. Recently published work by Goldy et al. (DOI: 10.1021/acsami.5c17072) describe elastic field power laws of vacancies in MXenes. These power laws extend to an energy environment around a vacancy. The energy environment provides theoretical backing for the observed distributions and clusters in Ti₃C₂T_x flakes. This reference has been added. Regarding concentrations, this is driven by processing conditions. Some defects are predicted by entropy, but these are modest compared to the observed quantities. We have added the following sentences to Results Section C in the main text.

Additional Text: “Material entropy predicts the presence of defects, but the harsh processing conditions drive the observed quantities of vacancies. The distribution of these vacancies is driven by the energy environment around the vacancies. Recent power laws for the elastic field of vacancies in MXenes describe this energy environment³⁸. Extending this environment allows us to isolate the key energetic drivers of vacancy formation and gain understanding of the energetic stability of defect clustering.”

5. *Regarding the statement “Overall, previous work shows that the etching of MAX phases to MXenes begins at the exterior surface of the grain³⁶, which would mean that the exterior flakes of MXene are exposed to the acid mixture for a longer duration than the interior flakes—this may serve to increase the number of vacancies in the exterior flakes. If this hypothesis is true, we believe this could suggest that the delaminated 5% HF Ti₃C₂T_x MXene flakes were delaminated from these etched MXene sheets nearer to the surface of the grain. While beyond the scope of this study, these results suggest that etching duration, in addition to etchant concentration, is a critical parameter for controlling 3D defect topology, warranting future investigation.” While I follow your logic and can accept the argument - that 5% HF samples are likely flakes closer to the exterior of the parent MAX grain (because the delamination was not complete) so they could/should have more vacancies, it is not clear what is exactly meant by this statement as the data shown in Fig. 2 seems to show that the 5% HF sample had the fewest point defects of the 3 samples. Can you clarify what is meant here?*

Our Response: We thank the Reviewer for their insightful question and the opportunity to clarify our work. We agree with the Reviewer’s point and the example of the 5% HF sample with a lower concentration of point defects compared to the 9.1% HF sample. To validate hypothesis, future works are certainly necessary. To this point, if cluster defects are potentially first formed at point defects by overetching reactions, and our 5% flakes are closer to the grain surface, both of which are hypotheses and not investigated in this work, this could also validate why the point defects are lower but the cluster defects are higher in the 5% sample compared to the 9.1%.

Additional Text: “Regardless, future studies investigating the role of flake locality and defect formation mechanisms based on the MAX grains are necessary to better test this potential explanation.”

Reviewer 3

The authors investigate vacancy concentration and distribution in MXenes by machine-learning guided analysis of scanning tunneling electron microscopy (STEM) images, complemented by vacancy clustering simulations. While the main findings of the analysis appear sensible, I get the impression that lot of the potential problems are swept under the rug. This is especially true for the first section of Results.

About STEM imaging:

- 1. It's not clear to me what the "stacked STEM images" are? I first thought it means that images are taken from the same location, but the stacked images in Supp. Figures 8-10 look very different.*

Our Response: We agree this wording was confusing. “Stacked STEM images” simply referred to the way multiple images (of different fields of view) are presented in Figure 1b,c as though they are stacked in 3D. We have removed the word “stacked” to avoid this confusion.

- 2. Actually, it's also never clearly stated that the imaged flakes are monolayers, although I presume that to be the case.*

Our Response: We thank the Reviewer for pointing this out and have added the following statements in the Main Text Results Section A and Supplementary Information Note 1 Section B, respectively, stating that they are indeed monolayer flakes.

Additional Text: “We imaged these samples using STEM to obtain high-angle angular dark-field (HAADF) images of monolayer flakes in each MXene sample.”

Additional Text: “One to two monolayer flakes were randomly chosen for atomic-resolution data collection and defect analysis.”

- 3. Do the authors track whether vacancies are formed during imaging?*

Our Response: This is an interesting question. We did not notice any new vacancy formation during image acquisition. We did, however, notice movement of some of the vacancies, despite the use of a low 60 kV accelerating voltage to minimize primary knock-on damage. We further minimized this effect by collecting images at very low beam currents (10-25 pA), but nonetheless raw image stacks (which were rigidly aligned and summed to produce the images for analysis) show some hopping of vacancies between neighboring sites. Our model determines where the most likely vacancy positions are (commonly amongst all aligned images). In future work, we'd like to develop models that work in low-signal-to-noise environments to capture the movement of vacancies.

To address this comment, we have added a figure in the supplementary information showing the model performance as a function of the number of rigidly aligned images.

Supplementary Figure 2: Model Performance as Number of Images Increases

FIG. S2. Rigid Alignment of Images. To achieve higher effective resolution, raw images were rigidly aligned and summed as described in Supplementary Note 1B. Model performance is shown as a function of the number of aligned images. Red dots indicate atomic positions found by the model. While slight atomic movements are observed across the raw frames, image stacking was necessary to achieve the performance shown in the bottom-right panel. Future work will explore models capable of operating directly on extremely low-signal-to-noise images, focusing on time-resolved analysis rather than rigidly aligned averaging and allowing us to investigate defect dynamics. Scale bars 1 nm.

4. *Are the images taken from flake interior or exterior, and does the defect concentration depend on the location?*

Our Response: All STEM images were acquired from the interior/middle of monolayer flakes, as opposed to near the edges. It is entirely possible that there is variation in the defect concentration close to the flake edges. Analysis of such variation would be an interesting follow up question. However, we felt that a comparison between etching conditions for MXene

preparation was more technologically valuable as an example case for our 3D atomistic defect classification.

5. *Is it possible to corroborate the vacancy or termination concentration trends via e.g. EDX/XPS measurements?*

Our Response: We thank the Reviewer for this important point. In this study, EDX characterizations are potentially difficult to corroborate missing Ti to at% values to compare with our ML identification, as the remainder elements (C, O, F) are all rather light elements and would certainly change based on defect concentration, which would make corroboration with exact at% values rather challenging. In addition, we did not have the capabilities to conduct atomic resolution resolved EDX measurements in this study, owing to the unavoidable beam damage induced during long EDX mapping over several minutes. In future studies it may be possible to resolve intercalated elements (Na, K, etc.) at defect sites using more efficient EDX detectors, though single-atom spectroscopy is still incredibly challenging.

XPS is potentially useful, and we have made use of this technique with recently published studies to understand changes in Tx and water adsorption in defect sites (DOI: 10.1021/acsmaterialslett.5c01186). However, XPS faces the same issues with quantification of at% values as EDX does, as if we assume Ti changes, C, O, F, OH could also change, which makes a stable comparative value complicated and based on potentially inaccurate assumptions. In addition, XPS lacks the resolution necessary in this work to validate our STEM imaging; therefore, we did not focus on this technique in this study.

To our knowledge so far, the only larger-scale technique which could accurately be used to validate these results is the layer-resolved secondary ion mass spectrometry (SIMS). However, even SIMS' layer-by-layer atomic concentration quantification lacks the ability to identify types of lateral defects (clusters, nanopores, etc) as we do in this work, as the lateral resolution is far above what STEM is capable of (few microns resolution in SIMS compared to single atoms in STEM). We consider this a major advancement, allowing STEM to quantify defects as accurately as SIMS and provide additional information quickly using ML techniques, benefiting the community.

Finally, our vacancy concentrations agree with prior manual STEM measurements (DOI: 10.1021/acsnano.6b05240). We have added the following supplementary figure corroborating the vacancy percentage values we found in this study.

Supplementary Figure 5: Vacancy Percentage Comparison

FIG. S6. Comparison of Vacancy Percentage to Past Study. The vacancy percentages found in this study are compared to a previous study³, which confirms the trend seen in our study. Values and error bars from the previous study are estimated using Plot Digitizer.

6. *No explanation is given to the obvious "striping" in all STEM images. Could it be related to the functional group distribution?*

Our Response: The “striping” in the STEM images is due to slight rippling in the MXene flakes. This effect was demonstrated in STEM images of monolayer $\text{Ti}_3\text{C}_2\text{T}_x$ flakes by Sang, et al. (ACS Nano 10, 9193–9200 (2016)).

Additional Text: “We note that many of the images exhibit some ‘striping’ in the MXene lattice. This is due to rippling in the MXene flakes causing slight local tilt of the c axis. The same effect was previously demonstrated by Sang et al.³”

About ML model:

7. *Little details are given in the main text about how the vacancy layers is determined. The authors can identify vacancies in 3 different layers, since Ti atom in each layer has different lateral position, but they cannot say which layer is which. The low concentration layer is always taken to be the middle layer. This may very well be correct, but it's nevertheless an assumption that is important to lay out in the main text. Consequently, claiming that a map of 3D topology is created is a bit questionable.*

Our Response: We agree with the Reviewer that more information on the determination of vacancy layers is needed in the main text, as was also stated by Reviewer 1. We have added text to resolve this issue, as described in our response to Reviewer 1.2.

8. *If I understood correctly, the ML model was trained using different crops from one microscopy image from an old paper. This sounds dubious, both for the small amount training data and the fact that all the experimental details are slightly different. Since the authors have a lot of own images, why not use those?*

Our Response: The Reviewer has pointed out a fundamental difficulty for training ML models in materials science, specifically the lack of training data, which is a topic our group has extensively published on (DOIs: 10.1038/s41524-021-00652-z; 10.1017/S1431927622012065; 10.48550/arXiv.2311.08585). We have added these references to the supplementary information. We appreciate them for pointing out that we did not fully explain the reasons behind our training methods. We have added additional text in Section C of Supplementary Note 1 expanding on this.

Additional Text: “Our own data was time-consuming to image, and we did not want to lose valuable information that could be utilized in our statistical analyses. In addition, we wanted to create a model that was robust to change across experimental settings, so it could be reused across a variety of MXene images. Each STEM instrument has its own environmental effects, and these are subject to change over time. Therefore, we would like to create a model that works generally for MXene images and is not specialized for one instrument’s data on a particular day. To accomplish this, we turned to a previous paper³, and used crops from one high resolution, large frame of view image to train our models.”

About the MC simulations:

9. *The authors use bond-order potential developed for MAX phase. Here it is applied to MXenes without any testing on its accuracy, in particular how it works for the terminations.*

Our Response: We thank the Reviewer for pointing out this inconsistency. The potential we used is specifically fit for MXene monolayers. It was based on the MAX potential from Plummer et al., but they extended it to MXenes. The correct reference has been added to the supplement (Supplementary Reference 8).

10. *The authors do not mention if there is a correlation between the Ti vacancy formation with the terminations in the neighboring sites?*

Our Response: We appreciate this point regarding higher order vacancy interactions. The Monte Carlo model does not explicitly prevent surface terminations from occupying Ti vacancy sites. However, the energetic relaxation ensures that there are no isolated surface terminations, both types of vacancies converge towards combined clusters. Additionally, we always model fewer Ti vacancies than surface vacancies to avoid isolation. As far as migration and substitution of species into vacancies, this is part of an upcoming modeling specific publication. We have added the following statement in Section D of Supplementary Note 1.

Additional Text: “The MC model does not explicitly prevent surface terminations from occupying Ti vacancy sites. However, the energetic relaxation ensures that there are no isolated surface terminations.”

11. *All that said, I believe the extracted vacancy positions are still largely valid and their clustering can then obviously be analyzed. This part of the manuscript provides clear trends on the types of vacancies/vacancy clusters and their dependence on the HF concentration, which should be of considerable interest to the community. However, since there are many technical details missing and the validity of the approaches insufficiently substantiated, I think the manuscript is not suitable for Nature Communications in the present form.*

Our Response: We thank the Reviewer for their assessment of the manuscript's novelty and believe we have clarified the necessary points.

Sincerely,

Steven R. Spurgeon, Ph.D.
Senior Materials Data Scientist
National Laboratory of the Rockies (NLR)

Joint Faculty
Metallurgical and Materials Engineering
Colorado School of Mines

Research Fellow
Renewable and Sustainable Energy Institute (RASEI)
University of Colorado Boulder

March 9, 2026

We thank the Editor and Reviewers for their informative and timely feedback on our manuscript, NCOMMS-25-85357-T: “Revealing the Hidden Third Dimension of Point Defects in Two-Dimensional MXenes.” We have addressed all reviewer comments and editorial requests in Author Checklist, and we believe the manuscript is now ready for publication.

Reviewer #1:

I thank manuscript authors for addressing my comments and revising manuscript accordingly. With these changes, I recommend the manuscript for publication in its current form.

Response:

We thank the reviewer for their comments in the first round of edits and are glad that we have revised accordingly.

Reviewer #2:

I thank the authors for responding to the comments of myself and the other reviewers. After considering their responses to our comments, I can easily recommend this manuscript for publication in Nature Communications.

This time I was able to view the source code and from a quick review of the repo page it is clear to me that I could easily clone the repository and engage with the code in a local environment.

We thank the reviewer for their helpful comments and for pointing out the issues accessing the source code. We are glad to have fixed these issues.

Reviewer #3:

The authors have revised the manuscript in response to comments and question from three Reviewers.

In my previous report, I concluded that the vacancy clustering analysis is likely valid and useful to the community, but due to missing experimental details and problems in the computational methodology, I thought the manuscript was not suitable for Nature Communications.

In the revised version, the authors have provided additional details and clarifications concerning

the experimental part.

One of my main criticisms was that the bond-order potential developed for MAX phases was used these MXene systems without testing. In fact, the potential was also valid for MXenes. It was just incorrectly described in the text and missing relevant reference.

The small amount of data used to train the ML model remains a deficiency, but overall I think the manuscript has improved enough to be suitable for publication in Nature Communications.

Our Response:

We thank the reviewer for pointing out the discrepancies with the bond-order potential in our first draft. We are happy to have fixed the issue and to now have the correct references.

Sincerely,

Steven R. Spurgeon, Ph.D.
Senior Materials Data Scientist
National Laboratory of the Rockies (NLR)

Joint Faculty
Metallurgical and Materials Engineering
Colorado School of Mines

Research Fellow
Renewable and Sustainable Energy Institute (RASEI)
University of Colorado Boulder